# Small-scale utilitarianism: High acceptance of utilitarian solutions to Trolley Problems among a horticultural population in Nicaragua

**Jeffrey Winking** [1]*, **Jeremy Koster** [2]

**1** Department of Anthropology, Texas A&M University, College Station, Texas, United States of America,
**2** Department of Anthropology, University of Cincinnati, Cincinnati, Ohio, United States of America

* jwinking@tamu.edu

## Abstract

Researchers often use moral dilemmas to investigate the specific factors that influence participants' judgments of the appropriateness of different actions. A common construction of such a dilemma is the Trolley Problem, which pits an obvious utilitarian solution against a common deontological dictum to not do harm to others. Cross-cultural studies have validated the robustness of numerous contextual biases, such as judging utilitarian decisions more negatively if they require contact with other individuals (contact bias), they force others to serve as a means to an end (means bias), and if they require direct action rather than inaction (omission bias). However, such cross-cultural research is largely limited to studies of industrialized, nation-state populations. Previous research has suggested that the more intimate community relationships that characterize small-scale populations might lead to important differences, such as an absence of an omission bias. Here we contribute to this literature by investigating perceptions of Trolley Problem solutions among a Mayangna/ Miskito community, a small-scale indigenous population in Nicaragua. Compared to previously sampled populations, the Mayangna/Miskito participants report higher levels of acceptance of utilitarian solutions and do not exhibit an omission bias. We also examine the justifications participants offered to explore how Mayangna/Miskito culture might influence moral judgments.

## Introduction

The Trolley Problem is a well-studied moral dilemma in which a runaway trolley speeds towards five unsuspecting victims. On a side track stands only one potential victim. Next to the tracks is a bystander who has the opportunity to pull a lever, which switches the trolley to the track with only one individual [1,2]. The problem pits an unmistakable utilitarian solution against a common deontological dictum to not actively do harm to others. Under this traditional version, approximately four-fifths of sampled participants agree that pulling the lever is the appropriate thing to do [3]. However, researchers have modified this original scenario into an array of related dilemmas, each including the same utilitarian equation of five deaths versus

**Data Availability Statement:** Data used in this paper can be downloaded here: https://osf.io/tr84j/.

**Funding:** This research was funded by the Max Planck Institute for Evolutionary Anthropology

(https://www.eva.mpg.de/) and the Department of Anthropology, Texas A&M University (https://liberalarts.tamu.edu/anthropology/). The funders had no role in study design, data collection and analysis, decision to publish, or preparation of the manuscript. There was no additional external funding received for this study.

**Competing interests:** The authors have declared that no competing interests exist.

one, but varying by subtle contextual factors [4]. Often, these contextual changes lead participants to reject the utilitarian solution. By studying these biases, researchers can isolate the factors that influence participants' perceptions of the acceptability of utilitarian actions and thus more accurately define humans' moral deontology.

In an early variant, instead of a lever, the bystander stands next to a large man on a pedestrian bridge suspended over the track. The bystander has the option to push the man from the bridge so that his body would stop the trolley. In this variant, only approximately half report that pushing the man is the right thing to do [3]. Previous research has identified three elements in this footbridge variant that reduce perceptions of permissibility. First, individuals tend to judge actions that involve physical touch as less permissible (*contact bias*) [5,6]. Second, scenarios are deemed less permissible when they require another individual to serve as a means to the end (i.e., the large man is *used* to stop the Trolley) (*means bias*) [4,7]. Finally, the third factor, which is evident in both the traditional and the footbridge scenario, is the fact that the bystander must *act* in order to bring about the utilitarian solution, as opposed to simply allowing for the utilitarian solution to come about naturally (*omission bias*) [8–11]. For instance, it would presumably be less offensive if the bystander simply refrained from warning the large man who was distracted and was about to walk off the footbridge and inadvertently stop the trolley from killing five men.

Studies of the biases that impact Trolley Problem decisions are often presented as if revealing fundamental elements of human moral psychology [3]. Indeed, there is a great deal of evidence for an overarching nature that defines human morality [12,13]. The *degree* of cross-cultural variation in moral reasoning is an empirical question, however, that can be revealed only through a broad cross-cultural record. Regarding studies of Trolley Problem scenarios, many have purportedly validated the cross-cultural robustness of the biases that have been detected [3,7]. For instance, in a recent online survey that included participants from 42 countries, Awad et al. (2020) [3] found that while there was a great deal of variation in overall acceptance rates across populations, the samples from all 42 countries exhibited both contact and means biases. However, the participant pools within these populations were limited to those who had access to a computer, the internet, and who were literate in a national language. This narrow sampling is a longstanding bias in social science research, although there is a growing number of researchers endeavoring to expand the breadth of cultural diversity included in such research [14–18]. The common inclusion requirements for social science research are likely to exclude participants from small-scale populations.

## Moral reasoning and small-scale society

Although there is no one defining element of small-scale populations, and while these populations represent a diverse menagerie of cultures that exist around the world, they share a number of social, demographic, and cultural elements that differentiate them from WEIRD (Western, Educated, Industrial, Rich, Democratic) populations [15,19]. Population centers tend to be much smaller and arranged by kinship. Societies therefore tend to be more close-knit, and day-to-day social life is dominated by interactions among individuals who know one another quite well. This familiarity facilitates communal labor, which tends to be more common than in WEIRD populations. Many small-scale societies have not gone through (or are in the process of going through) the demographic, epidemiological, and nutritional transitions that characterize the populations that social science research most often draws from. This means that these societies often exhibit higher fertility and mortality rates, that mortality is driven more by infectious than chronic disease, and that individuals in these societies tend to live on a subsistence-level with a more tenuous energetic balance.

Previous research suggests that the differences described above can influence moral reasoning in a variety of ways. Here, we explore three possibilities. First, according the dual process theory of moral reasoning, utilitarian moral reasoning is achieved by a deliberative, calculated, and cognitively costly manner of thinking, whereas deontological moral reasoning is governed by more automatic, emotional thinking [20,21]. Therefore, factors that facilitate costly and purposeful cognition tend to increase utilitarian reasoning, whereas those that hinder it are associated with deontological reasoning [22]. A number of cultural elements more common to small-scale societies are associated with deontological reasoning, including lower education levels [23], lower economic status [24], and higher norm conformity [25], suggesting that deontological reasoning might be more common in such societies.

Second, researchers have found that notions of social connectedness tend to induce utilitarian moral reasoning, with participants being more willing to act to sacrifice one in order to save five when the actors in the scenarios are imagined to be relatives or in-group members [11,26,27]. Similarly, Haidt and Baron [11] showed that, while participants exhibited the deontologically-motivated omission bias when considering strangers (judging ethically dubious *actions* more impermissible than ethically dubious *failures* to act), participants did *not* exhibit such an omission bias when judging the actions of friends. In one of only two studies to date of Trolley Problems conducted among small-scale populations, Abarbanell and Hauser [9] found that participants in a Maya community in Mexico also did not exhibit an omission bias (although they did exhibit a contact/means bias). Following the logic of Haidt and Baron, they reasoned that the higher connectedness among participants in small-scale societies might have lead Mayan participants to project such connectedness onto the actors in the scenarios, leading to the lack of the omission bias. Contrary to the previous paragraph, this line of research suggests that members of small-scale societies might be more oriented toward utilitarian moral reasoning, and that, in particular, omission biases might be less common.

Finally, there are ad hoc sociocultural factors whose influences are unknown because they are impossible to investigate among WEIRD populations or because they are simply unique to a particular culture. For instance, the experience of living at a subsistence level, with high rates of mortality, and with a more tenuous energy balance, cannot be replicated in the lab by having participants read a vignette. Under such circumstances, utilitarian moral reasoning might dominate by necessity, similar to how such reasoning takes precedence in medical triaging during large-scale crises [28].

For a number of cultural elements, which are unique to each population, the influences can only be elucidated through ethnographic inquiry. In the second study exploring Trolley Problems among a small-scale population, Sorokowski, et al. [29] reported that the Yali of Papua, Indonesia rarely chose the utilitarian option (to sacrifice a single man to save five), exhibiting a rate lower than any other previously reported [3]. They argue that the low rates are in large part due to participants' fears that the victim's kin would try to avenge his death, a practice that is common in Yali culture. Note that if that interpretation is true, the traditionally deontological solution (to not sacrifice the one man) is in reality not being chosen due to deontological reasoning, but due to a calculation of the consequences of that action. This highlights the importance of ethnographic inquiry in understanding the cultural norms and values that shape participants' interpretations of the options provided in the Trolley Problem scenarios. It is not possible to interpret results of such studies when researchers cannot discern which option represents the deontological versus the utilitarian decision.

Altogether, there are clearly a number of challenges to establishing straightforward hypotheses in the current study, particularly when so little research has been conducted among small-scale populations. The studies that do exist illustrate how the inclusion of small-scale populations often reveals novel patterns and previously unexplored cultural contexts, highlighting the

importance of expanding the exploration of moral reasoning beyond cultures of industrialized nation-states. Here we contribute to this literature by presenting a study of Trolley Problems among a small-scale population in Nicaragua. We include examinations of the classic switch/footbridge scenarios, as well as a study of omission bias in order to examine the degree to which such an effect is an artifact of Western populations. Importantly, we are the first to conduct a content analysis of the justifications for individuals' responses in such a population, which sheds a contextual light on the motivations that underlie the differences in responses in this population compared to those in previous studies.

## The Mayangna and Miskito population

Research took place in a village of approximately 350 indigenous Mayangna and Miskito horticulturalists in Nicaragua's Bosawas Biosphere Reserve. The Miskito and Mayangna are closely-related indigenous populations that reside in eastern Nicaragua and Honduras, which commonly intermarry and therefore share a sense of indigeneity that separates them from the larger Mestizo population [30]. The study village is culturally Mayangna, although a number of Miskito individuals have married into the community, which is composed of approximately 45 households.

The indigenous populations have a centuries-long history of interaction with the colonizing British. The Miskito populations interacted more closely with the British whereas Mayangna individuals frequently express pride that their culture was less impacted. Despite this, English loan words are omnipresent in both of the indigenous languages, such as "Plis" [Please], "Excus" [Excuse me], and "Want" [Want]. More recently, cultural shifts have been occurring due to increasing access to technology and government policies/programs that are now reaching the village. Around five resident adults are employed by the government as schoolteachers in the under-sized two-room school building. Enrollment in the school is universal among children and a daily meal of rice and beans is supplied by the government. While literacy is now quite high among older children and young adults, it is quite variable among older individuals. Approximately three households presently own boat motors and chainsaws, and several more now have solar panels. In 2015, cell-phone towers were constructed along the Coco River and cell service now reaches within a three-hour walk from the community.

The population throughout this area also has a long association with the Christian faith. Although the Moravian Church has a longer history and a larger reach in the Mayangna population, nearly all members of the research community are Catholic. Church services delivered in the Mayangna language are regularly attended by community members at the recently constructed church. However, a number of beliefs and practices deviate from modern doctrine. These include the belief that individuals can be possessed by demons, the devil, or evil forces, as well as the practice of social marriage—living together and having children—for a number of years before, if ever, pursuing a formal religious marriage.

Despite the numerous changes, many aspects of life for this Mayangna village are likely very similar to those of previous generations (Conzemius 1932). The community is largely oriented around the nuclear family, with related families often building their houses near one another. Marriage is nearly universal by the early twenties and fertility is high, with total fertility rates exceeding seven births per woman [31]. Most of the food consumed by households is directly produced. Families maintain agricultural fields in which they grow rice, beans, and other crops, and also keep chickens, pigs, and, for many households, cows. They supplement this with earnings from the selling of produce and livestock, as well as panning for gold.

**Mayangna/Miskito culture and moral reasoning.** There are a number of cultural features in this community that likely impact how individuals perceive moral dilemmas compared to how they are viewed in Western populations. For instance, like in many small-scale

communities, there is a greater collective identity and motivation compared to Western populations. Private ownership is widely recognized, although inter-household food sharing is relatively common, particularly for meat [32]. The collective nature of the community is best illustrated by regular communal labor. This includes labor tasks that benefit the community at large, such as clearing the soccer field, renovating the church floor, or constructing a communal granary. The tasks are organized by the village leaders and by the community at large in community-wide meetings, and households are typically expected to contribute comparable amounts of labor. For large projects, such as the construction of a church, formal schedules are created with teams working in multiple-day shifts. When individuals are unable or unwilling to contribute their expected labor, they must pay a fine in money or food.

While community work is frequently performed without pay, when laborers are required for individual projects, such as the clearing of a new field or the construction a house, the beneficiary is typically expected to pay the laborers a daily wage. This can even include occasions in which a man helps his brother, although this depends on the agreement between the two. These exchanges extend to urgent situations as well. For instance, after a cow died of a snake bite, the owner had to pay his brother to help him bury it.

Another feature which might influence perceptions of moral dilemmas is the greater familiarity with death compared to Western contexts. Reproductive histories of community members and their parents suggest a child (under-five) mortality rate of approximately 96 per 1,000 live births [15]. In the U.S., this figure is 6.5 [33]. Furthermore, many of the older men took part in the wars in the 1980s, fighting as Contras against the Sandinista army.

The picture that emerges is a close-knit community, which, like many small-scale populations, exhibits a high level of community-level self-reliance and interdependence, reduced cultural and religious diversity, and uniformity in life histories, including relatively high mortality. As described in the previous section, it is not clear how moral reasoning might be influenced by the numerous sociocultural elements that differentiate this population from Western contexts. It is the authors' impressions that moral reasoning by individuals in this population is guided more by community- and religion-based frames [34], and that there is a greater expectation of norm conformity. Education rates are increasing but still lower than Western contexts, and most live close to a subsistence level of existence. All of these elements might suggest an orientation toward deontological reasoning. However, the close relatedness among inhabitants, as well as living closer to the edge of potential health and economic calamity might drive a greater reliance on utilitarian moral reasoning.

Finally, as also argued previously, it is possible that local interpretations of utility and deontology do not align with Western understandings. Utility can only be gauged by knowing how positively and negatively different outcomes are perceived by members of a population. While the revenge killings might be less common in Mayangna and Miskito populations compared to the Yali population, justice is frequently pursued on individual and community levels, and revenge killings do occur. Furthermore, cultures likely vary in how wrong they perceive directly causing a person's death to be. For example, cultures with a more fatalistic orientation might be more likely accept the death of the five men [35]. While, we have no inclinations as to how the Mayangna and Miskito might perceive such acts, we explore this below in our content analysis of spontaneously offered justifications.

## Materials and methods

### Procedure

Two separate studies took place, one in the winter of 2018 and the other in the summer of 2019. The first study served in part as a pilot study to validate that the scenarios and task could

be well understood by participants (previous experiences with comparable projects have met with mixed success in this population). Study 1 was thus simpler, whereas Study 2 was more in-depth, following the procedures of Abarbanell and Hauser [9], and included a content analysis of comments that were spontaneously offered.

**Study 1.** The first study is based on the more-common forced-choice design, in which participants respond as to how they would act when presented with a Trolley Problem scenario (e.g., to pull the lever or not). Four Trolley Problem vignettes were translated into Mayangna by two translators who are fluent in Spanish and Mayangna (see the Supporting information for translations). All adults in the community were invited to the project house to participate in interviews relating to social transactions (for an unrelated study), and the Trolley Problem portion, which included a single Trolley Problem vignette, was administered at the conclusion of these interviews.

For each interviewee, one of the four vignettes was randomly selected (Fig 1, first four scenarios). The vignettes included the traditional Trolley Problem scenario involving the switch (Trolley Switch), as well as the scenario involving the large man on a bridge (Trolley Push). In order to provide more culturally salient versions, we recreated these scenarios involving a stampede of cattle in which a man had the opportunity to 1) push a gate closed that directed the stampede away from five men and toward one man (Stampede Gate), and 2) push a man sitting on a fence into the opening in the fence, thereby killing the man, stopping the stampede, and saving five men beyond the opening (Stampede Push). Participants were shown a picture of the scenario and read the accompanying vignette (for four individuals who were not fluent in Mayangna, the vignettes were translated verbally into Miskito). Participants were then asked if they would take the necessary action. All interviews were conducted by a local research assistant fluent in all three local languages. Combined with the other interviews, the process took approximately 45 minutes and participants received 60 Cordobas (approximately US$2).

The research design (for the unrelated study) included multiple rounds of interviews, and 46 individuals ended up responding to two Trolley Problem vignettes. For these participants, the respective vignettes were administered and evaluated on separate days, usually at least one week apart. As the scenarios were selected at random, 17 individuals were presented with the same scenario twice, in which case we retained only their first response for the analysis.

Ninety-five individuals participated in the first study (approximately three quarters of the adult population). After removing duplicates, a total of 32 participants responded to the Trolley Switch and Trolley Push scenarios each, 29 responded to the Stampede Gate scenario and 33 to the Stampede Push scenario. These sample sizes allow for a power of 0.73 and 0.72 ($\alpha$ = 0.05) to detect contact/means bias in the traditional and cultural salient versions, respectively, assuming the population exhibited the global proportions reported in Awad et al. (2020) [3] (0.81 opting to pull the lever and 0.51 to push the man).

**Study 2.** For comparability, the procedures of the second study follow those employed by Abarbenell and Hauser [9]. The primary differences from the first study were the inclusion of four additional scenarios to test an omission bias, the use of a scale for participants to rate the acceptability of the utilitarian option (in lieu of asking how they would act), and a content analysis of justifications that were spontaneously offered. Following Abarbenell and Hauser [9], this study only evaluates ratings of utilitarian decisions, and so it does not speak to perceptions of deontological decisions.

Eight Trolley Problem vignettes were translated into Mayangna by the same two translators (Supporting Information). They were then back translated into Spanish by a third translator. The few differences were then discussed and changes were agreed upon. Participants were invited to come to the project house where the interviews took place.

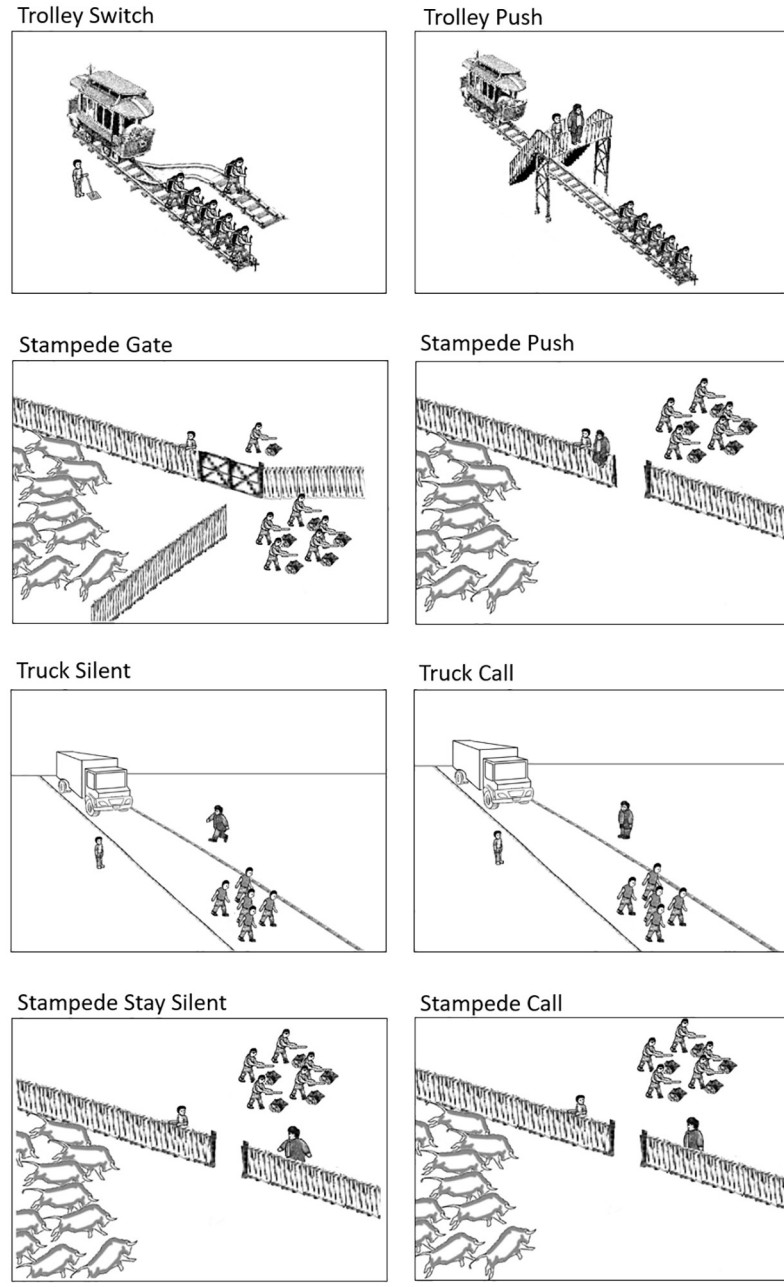

**Fig 1. Trolley Problem scenarios (modified from Hauser et al., 2007)** [7].

Following Abarbanell and Hauser [9], the second project aimed to explore individuals' perceptions of the appropriateness of the utilitarian decisions. Participants were presented with a visual 5-point scale which ranged from "Very Bad" ("Uba Dutni") to "Very Good" ("Uba Yamni"). These dichotomous phrases were deemed to be optimal anchors as they are frequently spoken and represent the major bad/good spectrum that is regularly used in the community. The three other options on the scale were "Bad" ("Dutni"), "Neither good nor bad, but in the middle" ("Yamni awasa, dutni awasa, papaskat"), and "Good" ("Yamni"). Faces exhibiting frowns and smiles accompanied the large icons and text to help those who were sub- or

illiterate. Two examples, one exhibiting a clear good action (bringing food to a sick individual) and one exhibiting a clear bad action (stealing money from someone's house), were provided using the scale, and participants were judged proficient if they judged both in the expected direction. The vignettes were accompanied by pictures (Fig 1) and were read in Mayangna (four individuals who were more fluent in Spanish preferred them to be read in the original Spanish), and participants were asked if they understood the vignette and if they had any questions. They were then asked what they thought of the man's decision (which was always the utilitarian solution). All interviews were conducted by JW and the same research assistant from Study 1.

The vignettes included the four versions in the first study as well four additional vignettes aimed at exploring the omission bias (Supporting Information). Following Abarbanell and Hauser [9], two scenarios describe a truck speeding toward a group of five pedestrians and a bystander who has the opportunity to 1) refrain from cautioning a man who was already about to walk into the truck's path (Truck Silent), or 2) call a man on the other side of the road to walk into the truck's path (Truck Call). Again, to provide a more culturally salient option, we included scenarios in which a stampede of cattle was running toward a group of five men beyond an opening in the fence and the bystander had the opportunity to 1) refrain from cautioning a man from walking into the stampede's path in the fence opening (Stampede Silent), or 2) call a man over toward the fence opening and in the stampede's path (Stampede Call).

Participants were presented with four scenarios. These consisted of one scenario from each of the four pairs of scenarios: 1) Trolley Switch or Trolley Push; 2) Stampede Gate or Stampede Push; 3) Truck Silent or Truck Call; 4) Stampede Silent or Stampede Call. The order of the four pairs was randomized, as was which scenario of each pair was included. This was done for three participants, and then for the next three, the reciprocal of the preceding three was used in order to ensure balance. Order was again randomized. The interviews took approximately 20 to 30 minutes and participants received 50 Cordobas (approximately $1.50).

For Study 2, 120 individuals participated, which included all available adults in the community, resulting in sample sizes ranging from 59 to 61 for each scenario. These sample sizes allow for a power of $>0.99$ ($\alpha = 0.05$) to detect a contact/means bias equal to the large effect (Cohen's d = 2.1) reported in Hauser, et al. [36], and a power of 0.9 to detect a medium effect size of Cohen's d = 0.6. Additionally, the sample sizes allow for a power of 0.86 ($\alpha = 0.05$) to detect the medium omission bias (Cohen's d = 0.5) equal to the median effect size among 11 scenarios across two studies exploring omission biases.

While we strived for standardization, the reading of long, strange stories involving extraordinary circumstances and foreign objects naturally led to numerous questions by participants. Members of this community are accustomed to such peculiar methods due to years of anthropological research conducted by the authors. However, conversational interaction is still much more culturally familiar than reading (either aloud or to oneself) when compared to Western contexts. This presents the familiar tradeoff in cross-cultural research between standardization and comprehension [17]. The research assistant was trained to answer the questions in ways that were least likely to evoke notions of rightness or wrongness. For instance, he would answer questions using the phrases "one would die" and "five would not die" instead of saying "killed" or "saved."

*Content analysis of justifications.* Any spontaneous comments made during the rating process in Study 2 were translated by the research assistant and recorded by JW. These were subsequently coded by JW and a graduate student using three codes—*Utilitarian*: those offering a utilitarian rationalization (such as "It's better for one to die than five."), *Deontological*: those offering a deontological principle/rationalization (such as, "This man also has a right to live."), and *Tradeoff*: those acknowledging a tradeoff between the options (such as,

"It's wrong to kill a man, but saving five is better."). These codes were chosen as they represent the two modes of moral reasoning that conflict in Trolley Problems (utilitarian vs. deontological reasoning), and the third code represents the conflict itself. Each comment could be assigned any combination of the codes. All coding was done after data collection and coders were blind to the participants' ratings. There was sufficient agreement in the coding of *Utilitarian* (Cohen's kappa = 0.75) and *Tradeoff* (Cohen's kappa = 0.75) [37]. However, relatively weak agreement was evident for *Deontological* (Cohen's kappa = 0.41), and only baseline frequencies for this code are provided below. Only comments that were coded positively by both coders are included in analyses below.

### Ethics statement

This research was conducted under Texas A&M IRB Protocol IRB2019-0465D, which was approved for this study.

## Results

### Study 1

The first study includes responses to a total of 146 scenarios by 95 unique individuals, including 48 women and 47 men. As can be seen in Fig 2, the contact/means biases are apparent when asking how individuals would act in these situations. The effect exhibited in the Trolley scenarios is in the predicted direction but is not significant (Trolley Switch: 93.5%, Trolley Push: 78.1%, Fisher's Exact, $p = 0.148$), while the effect is much more pronounced in the Stampede scenarios (Stampede Gate: 93.9%, Stampede Push: 62.1%, Fisher's Exact, $p = 0.003$). These result in an odds ratio of 4.06 and 9.47, respectively. In both versions, the proportions responding that they would push the individual are higher than the 50% that are reported in a recent large-scale cross-cultural study [3]. The rates reported here would exceed those of all 42 countries for scenarios involving the mechanisms (switch/gate) and for those involving pushing a man, save for one comparison (the Trolley Push acceptance rate in Vietnam is slightly higher than that reported here for the Stampede Push scenario) (ibid.).

Post-hoc multivariate analyses including age and gender reveal a modest gender effect across the Trolley scenarios, with men slightly more willing to engage the utilitarian solution,

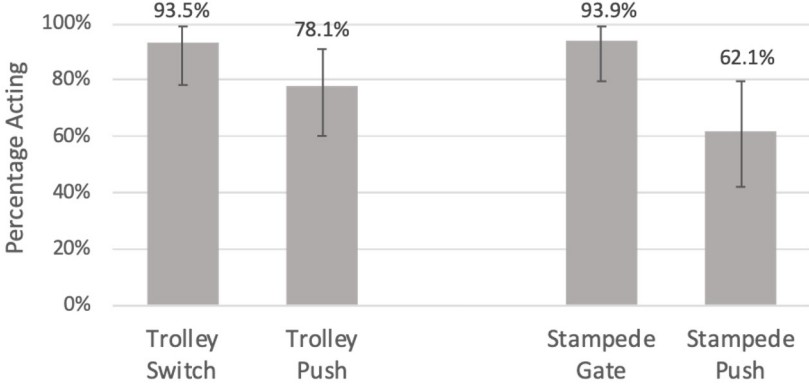

**Fig 2. Proportion of participants saying they would act to save the five.** No Contact/No Means, Trolley, n=32; Contact/Means, Trolley, n=32; No Contact/No Means, Stampede, n=33, Contact/Means, Stampede, n=29. Error bars are Clopper-Pearson exact 95% confidence intervals.

**Table 1. Logistic regression models of factors associated with choosing the utilitarian option.**

|  | *Estimate* | St. Error | *p* |
|---|---|---|---|
| Trolley Scenarios[a] |  |  |  |
| Intercept | 0.02 | 1.53 | 0.99 |
| Push Scenario = Yes | -1.13 | 1.37 | 0.41 |
| Gender = Male | 2.63 | 1.56 | 0.09 |
| Age | 0.01 | 0.04 | 0.73 |
| Stampede Scenarios[g] |  |  |  |
| Intercept | 5.27 | 1.65 | <0.01 |
| Push Scenario = Yes | -2.86 | 0.95 | <0.01 |
| Gender = Male | 2.54 | 1.03 | 0.01 |
| Age | -0.08 | 0.04 | 0.03 |
| Full Model[c] |  |  |  |
| Intercept | 4.03 | 1.21 | <0.01 |
| Push Scenario = Yes | -2.12 | 0.64 | <0.01 |
| Stampede Scenario = Yes | -0.73 | 0.56 | 0.19 |
| Gender = Male | 2.16 | 0.68 | <0.01 |
| Age | -0.04 | 0.02 | 0.11 |

[a]n = 63, unique individuals = 57. Random effect = ID.

[b]n = 62, unique individuals = 58. Fixed effects only due to limited variance in the random effect, ID.

[c]n = 125, unique individuals = 98. Fixed effects only due to limited variance in the random effect, ID.

and a significant gender and age effect across the stampede scenarios, with men and younger individuals being more likely to engage the utilitarian solution (Table 1). In a test including all four scenarios.

(Full Model), the inclusion of a contact/means component and gender remain significant predictors, whereas age and Trolley vs. Stampede (i.e., traditional vs. culturally salient) do not (Table 1).

Finally, as responses were collected in two separate rounds (with a number of individuals participating in both rounds), we compared responses in the two rounds as a measure of internal reliability (Supporting Information, S1-S3 Tables). This test revealed significant differences, suggesting doubt in the measure (Supporting Information S1 Table). Looking only at individuals who participated in both rounds (n = 47), the percentage of individuals not acting increased from 8.5% to 27.7% across all four scenarios. Explanations for this decline are necessarily speculative, but potentially relate to other aspects of the unrelated interviews that were conducted prior to the Trolley study. However, limiting the dataset to participants' first responses did not change the direction of the effect or the overall interpretation of the results, except that the difference between the Trolley Scenarios becomes more apparent (Exact Test, p = 0.097) (Supporting Information S2 Table).

## Study 2

In the second study, 120 individuals participated, including 62 women and 58 men. While there was a possible contact/means effect in the Stampede version (Cohen's $d = 0.24$, $p = 0.19$), the Trolley version showed no such difference ($d = 0.06$, $p = 0.72$) (Fig 3). In both tests of the omission bias, there were no discernible effects and the differences were in the opposite direction than predicted (Truck: $d = 0.08$, $p = 0.66$; Stampede: $d = 0.15$, $p = 0.40$) (Table 2). Post-hoc

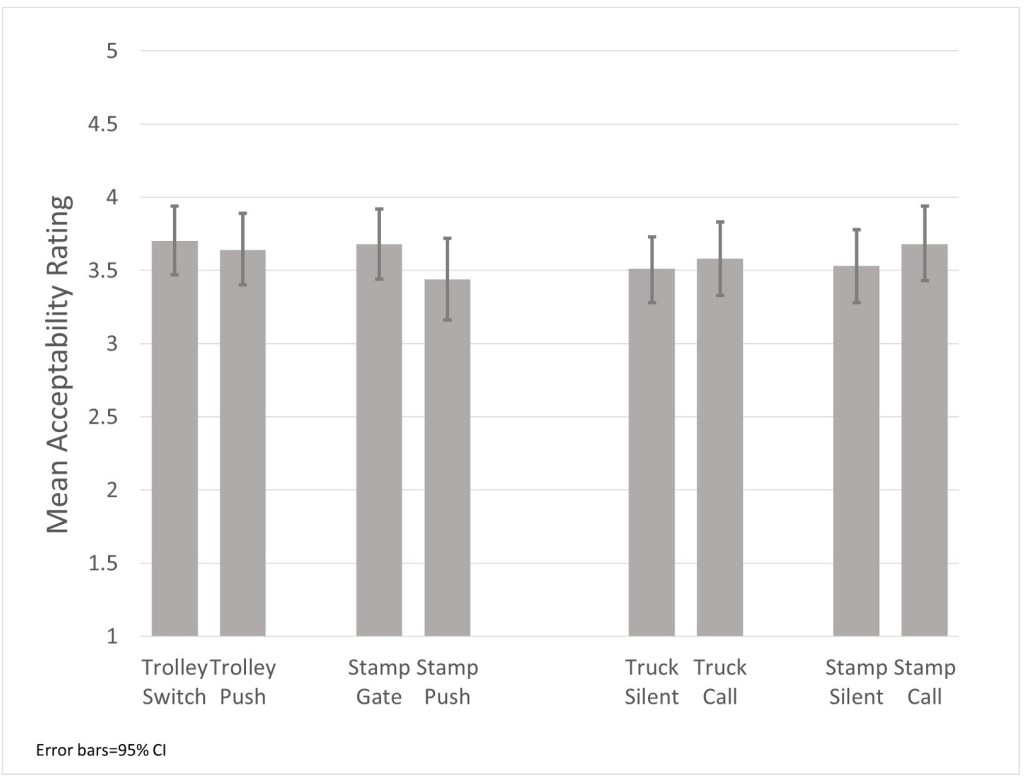

**Fig 3. Mean ratings of acceptability of utilitarian decisions across Trolley scenarios.**

analyses reveal no effects of gender or age and that the responses to the culturally salient scenarios were not significantly different from those to the traditional scenarios (Table 3).

Similar to Study 1, there is a remarkably high level of acceptance of the utilitarian solutions. Rates of rejection, those rating the action as Bad or Very Bad, ranged from a minimum of 10.3% for the Trolley Switch scenario to a maximum of 21.9% for the Stampede Push scenario. In every scenario, at least 60% judged the action to be Good or Very Good.

In a robustness check, we examined the possibility of order effects (Supporting Information, S4a-S4e Table). This supplementary analysis revealed a moderate effect with individuals reporting lower ratings for the first scenario. However, the inferences for the comparisons

**Table 2. Comparisons of the acceptability of utilitarian decisions across Trolley scenarios.**

|  | *n* | **Mean**[a] | *SD* | **95% CI** | *d* | *p* |
|---|---|---|---|---|---|---|
| Trolley Switch | 61 | 3.70 | 0.92 | 3.47–3.94 | 0.06 | 0.721 |
| Trolley Push | 59 | 3.64 | 0.94 | 3.40–3.89 |  |  |
| Stampede Gate | 60 | 3.68 | 0.93 | 3.44–3.92 | 0.24 | 0.193 |
| Stampede Push | 61 | 3.44 | 1.09 | 3.16–3.72 |  |  |
| Truck Silent | 59 | 3.51 | 0.86 | 3.28–3.73 | 0.08 | 0.655 |
| Truck Call | 60 | 3.58 | 0.96 | 3.33–3.83 |  |  |
| Stampede Silent | 60 | 3.53 | 0.97 | 3.28–3.78 | 0.15 | 0.401 |
| Stampede Call | 60 | 3.68 | 0.98 | 3.43–3.94 |  |  |

[a]Ranging from 1 = "Very Bad" to 5 = "Very Good".

**Table 3. Mixed effects linear regression models of factors associated with acceptability of utilitarian decisions.**

|  | *Estimate* | St. Error | *p* |
|---|---|---|---|
| Contact/Means Bias |  |  |  |
| Intercept | 3.94 | 0.24 | <0.01 |
| Push Scenario = Yes | -0.15 | 0.11 | 0.18 |
| Stampede Scenario = Yes | -0.11 | 0.10 | 0.28 |
| Gender = Male | -0.17 | 0.15 | 0.27 |
| Age | 0.00 | 0.01 | 0.60 |
| Omission Bias |  |  |  |
| Intercept | 3.26 | 0.25 | <0.01 |
| Call Scenario = Yes | 0.13 | 0.09 | 0.14 |
| Stampede Scenario = Yes | 0.07 | 0.07 | 0.29 |
| Gender = Male | -0.24 | 0.16 | 0.14 |
| Age | 0.01 | 0.01 | 0.14 |

[a]n = 228, unique individuals = 114.
[b]n = 228, unique individuals = 114.

between the scenarios remain consistent. The most substantive change was that there the non-significant effect between Stampede Gate and Stampede Push (Table 2, p = 0.193) becomes more pronounced in a mixed-effects regression controlling for whether or not scenarios were presented in the first position (Stampede Gate, EM Mean = 3.64, Stampede Push, EM Mean = 3.40; p = 0.085) (Supporting Information, S4e Table).

**Content analysis of comments.** While comments and justifications were not requested, 85.8% of participants offered at least one spontaneous comment while communicating a rating. These comments helped to ensure their understanding of the scenarios and to determine the reasoning behind their ratings. Table 4 displays the proportion of comments for each type by scenario. Participants spontaneously offered a comment including utilitarian logic in 39.8% of the ratings, accounting for 72.6% of all comments. Frequencies of utilitarian comments did not significantly differ across scenarios (n = 480, $\chi^2$ = 9.65, p = 0.21). Utilitarian justifications included comments like, "He did well, because only one will die instead of five," (in response to Stampede Push), and "Because he saved more lives, it's good," (in response to Stampede Silent). Many participants similarly commented on how bad the alternative would be, such as, "For five to die would be very bad," (in response to Truck Call), and, "His decision was very good; if he did not do that, five would have been killed," (in response to Stampede Gate). A

**Table 4. Percentage of participants offering spontaneous comments during ratings, by comment category and scenario.**

|  | Utilitarian | Deontological | Tradeoff |
|---|---|---|---|
| Trolley Switch | 42.6 (%) | 0.0 | 11.5 |
| Trolley Push | 33.9 | 0.0 | 11.9 |
| Stampede Gate | 35.0 | 0.0 | 10.0 |
| Stampede Push | 39.3 | 3.3 | 13.1 |
| Truck Silent | 35.6 | 3.4 | 3.4 |
| Truck Call | 56.7 | 1.7 | 6.7 |
| Stampede Silent | 40.0 | 0.0 | 5.0 |
| Stampede Call | 35.0 | 0.0 | 0.0 |
| Total | 39.8 | 1.0 | 7.7 |

number even praised the man, saying "If he does this, he is brave, and he kills one and not the five," (in response to Trolley Push), and, "The man is very intelligent, because if he screams, they would not hear; for this reason he is very smart, and it is good," (in response to Stampede Push).

Furthermore, utilitarian justifications were offered after 50.0% of the 310 positive ratings ("Good" or "Very Good"), compared to 10.0% of the 80 negative ratings ("Bad" or "Very Bad") ($n = 390$, $\chi^2 = 47.98$, $p<0.01$). Of the eight who rated the decision negatively *and* offered a utilitarian justification, six (75.0%) also referenced the tradeoff, compared to only ten (6.45%) of those who rated the decision positively and offered a utilitarian justification ($n = 390$, $\chi^2 = 84.08$, $p<0.01$). For instance, one participant reported, "One died so it's bad, although five are saved; so it's not that bad, but it's still bad," (in response to Stampede Push).

Although high agreement between coders was not evident for the code "Deontological," both coders assigned "Deontological" to fewer than 4% of comments, suggesting that deontological logic is infrequently employed. The five comments that both coders marked as including deontological reasoning included statements relating to general principles like, "This man also has a right to live," (in response to Truck Call), as well as to the specific actions, such as, "[It's bad] Because he pushed him."

In addition to the stated justifications, the foci of comments also shed light on the utilitarian reasoning. Overall, there was no significant difference in the frequency in which the lone potential victim was included in comments (mentioned after 39.4% of all ratings) compared to the group of five (40.8%). Comments referencing the lone individual did not differ in frequency among those who reported a positive ratings (40.1% commenting on the lone individual) versus negative ratings (36.5%, $n = 390$, $\chi^2 = 0.51$, $p = 0.47$). However, those reporting a positive rating were much more likely to offer a comment referencing the group of five (48.1%) than those reporting a negative rating (14.8%, $n = 390$, $\chi^2 = 28.68$, $p<0.01$). This again supports the interpretation of utilitarian reasoning, as those who focused more on the benefits to the group of five were more likely to endorse the man's action in the various scenarios.

## Discussion

We set out to explore how participants from a small-scale Mayangna/Miskito population responded to Trolley Problem scenarios, and in particular whether they exhibited the contact/means and omission biases reported in the literature for participant populations living in industrialized nation-state populations. Previous research had suggested that individuals in small-scale populations might report higher frequencies of deontological decisions [23–25], higher frequencies of utilitarian decisions [26], and lack an omission bias [9]. We also highlighted the need for more culturally contextualized studies, as individuals from different cultures might perceive the Trolley Problems differently, and they might follow different cultural norms that inform their decisions.

Regarding the biases, responses to the vignettes suggest that this population likely exhibits a contact/means bias but does not exhibit an omission bias. The two forced-choice scenarios in Study 1 resulted in a substantial effect for the culturally salient scenarios (OR = 9.47, p = 0.003), and a non-significant effect that was in the predicted direction for the traditional scenarios (OR = 4.06, p = 0.148). In the higher-powered study using moral judgments on an ordinal scale, neither the traditional scenarios (Cohen's d = 0.06, p = 0.721), nor the culturally salient scenarios revealed a significant contact/means bias (Cohen's d = 0.24, p = 0.193). However, both were in the predicted direction, and a post-hoc test which controlled for an order effect revealed a possible bias for the culturally salient scenarios (Difference of EM Means = 0.24, p = 0.085).

The two tests of an omission bias in Study 2 revealed no sizeable effect, exhibiting slight non-significant differences in the opposite direction than predicted. Haidt and Baron (1996) [11] showed that the omission bias was reduced when interactions occurred between friends—all infractions were judged comparably bad. Abarbanell and Hauser (2010) [9] suggested that such an effect might explain the absence of an omission bias among the Mayan population in their study, as individuals might project the higher level of interconnectedness of Mayan society onto the interactants in the scenario. This would suggest that omission biases are generally less common in small-scale societies, and we indeed failed to detect one in the current study. However, Haidt and Baron (1996) and Abarbanell and Hauser (2010) [9,11] found that participants rated utilitarian decisions as equally *bad*; among this Mayangna/Miskito community, they were rated equally *good*.

Concerning overall orientations toward utilitarian or deontological reasoning, individuals in this community report a much higher level of acceptance of utilitarian solutions to a range of Trolley Problem scenarios compared to previous studies. In all four scenarios of the initial pilot study (Study 1), participants opted for the utilitarian solution more frequently than that reported in all 42 national samples included in the Awad et al. [3] study, save for one case. Study 2 replicated this result. Acceptance rates were substantially higher than those reported in the study by Abarbanell et al. (2010) [9] (on which Study 2 was based).

It is possible that the effects in Study 2 represent an acquiescence bias—a general tendency to express agreement with survey questions. However, this concern is allayed by the fact that Study 1 revealed the same utilitarian tendency, despite nothing leading participants to choose the utilitarian option. Furthermore, in previous studies, members of this community were consistently willing to offer negative moral judgments of other types of hypothetical scenarios [18]. And finally, the comments that were spontaneously offered in Study 2 overwhelmingly employed utilitarian logic and did so in a way that suggests they are meaningfully linked to the evaluations—i.e., participants viewed the man's actions as a utilitarian solution, and they agreed with it. There appears to be little ambiguity in interpretation of the scenarios in this community, unlike that reported in the Yali study [29]. In that study, it is possible that participants viewed letting the five men die as the utilitarian solution (or at least the less personally dangerous solution), as it would be less likely to initiate revenge killings [29].

A likely explanation for the very high endorsement of utilitarian choices in this study is that such solutions are simply deemed more acceptable among this population compared to previously sampled populations. While we are reluctant to proffer why this might be the case, the comments that were offered provide evidence that the moral reasoning used by these participants included considerations that are likely not taken into account in WEIRD populations. Many of these relate to the considerations of stakeholders beyond the men in the scenario, supporting previous observations that moral reasoning by individuals in non-WEIRD populations relies more on community-based moral frames [34,38]. For example, one participant commented that, "If five die, it is a great cost to the community. It's a lot of work to bury all of them. One is much less." Another reported, "The Mayangna are few in number. If five die, that's a lot. If he calls him, he saves five. It's good because he's thinking of the five." Others discussed the cost to the families, such as, "There's always trouble. If five die, there will be many children who will be abandoned. He was thinking of the five."

Other comments illustrate how some participants related the scenarios to their experiences with mortality, experiences which are likely very different from those of industrialized populations. One individual reported, "It's better that one dies instead of five, like in a war, when someone is trying to save a group of kids." And another commented that, "In this culture, if you kill someone, his family will try to kill you. But he still saved the five." While this does

occasionally occur, it seems less common a cultural practice than that exhibited by the Yali, and was only mentioned by a single elderly participant [29].

Compared to Western contexts, it is likely that individuals in this population encounter more frequent scenarios in which decisions must be made with potentially grave consequences. Illnesses and injuries are more common, and when they occur, families must decide if a multi-day trek to a hospital is warranted; individuals must often personally engage to deal with violence, or even murders, that occurs within their communities; teams of armed men must travel to nearby homesteads to escort non-indigenous squatters from the protected reserve. Furthermore, such events occur within the framework of a subsistence-level existence in which there is often less of a margin to endure misfortune. Under such circumstances, it is possible that utilitarian reasoning simply becomes more of a necessity.

A follow-up focus group of three men and one woman reinforced the utilitarian motivations. They agreed that it was always important choose the solution that saved the most lives, even if the group to be saved included only two men instead of five. They were unable to offer any possible reasons why people in their community were more accepting of utilitarian decisions compared to people in other cultures. Such logic is not entirely impervious to local moral codes, however. They strongly agreed that they would *not* push the gate to save the five if the lone individual on the other side were a woman—"It is not good to see a dead woman," one man explained.

## Conclusion

While there may be some basic overarching precepts that define all of human moral reasoning, researchers are likely to overestimate these precepts when they limit investigations to industrialized populations. For example, we have provided evidence here that individuals in a Mayayngna/Miskito population likely exhibit a contact/means bias, but no omission bias. More strikingly, the results suggest that they exhibit higher rates of acceptance of utilitarian solutions to Trolley Problems than all populations included in a large cross-cultural study [3]. However, Sorokowski, et al. [29] found that participants in a small-scale population in Papua, Indonesia, exhibited the diametrically opposite tendency—they exhibited *lower* levels of acting to sacrifice the lone man to save the five. Thus, the remarkable variation exhibited in Awad et al.'s sample would have been expanded even further by each of the two most recent studies working with small-scale populations. And while these two groups might share a number of characteristics in relation to Western populations, they also both exhibit unique cultural elements that drive these differences.

For cross-cultural research, however, it is not enough to rely on simplistic dichotomies of WEIRD versus non-WEIRD, or small-scale versus non-small-scale. The diverse cultures of the world vary on multiple dimensions of social and political organization, demography, and economic orientation that plausibly interact with an evolved psychology to generate substantial cross-cultural heterogeneity in moral reasoning. It would be beneficial to have theoretical models that can elucidate the underpinnings of cultural variation in utilitarian norms and related biases.

## Supporting information

**S1 File.**
(DOCX)

## Acknowledgments

We thank the Mayangna and Miskito communities with which we worked. We also thank Catharina Laporte for early conversations about the topic, Leonie Ette for her assistance in data collection, and Angela Achorn for coding comments.

## Author Contributions

**Conceptualization:** Jeffrey Winking, Jeremy Koster.

**Formal analysis:** Jeffrey Winking, Jeremy Koster.

**Methodology:** Jeffrey Winking, Jeremy Koster.

**Writing – original draft:** Jeffrey Winking.

**Writing – review & editing:** Jeremy Koster.

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
