## [Decision Letter · Decision Letter 0]

13 Jan 2021

PONE-D-20-35491

Small-Scale Utilitarianism: High Acceptance of Utilitarian Solutions to Trolley Problems among a Horticultural Population in Nicaragua

PLOS ONE

Dear Dr. Winking,

Thank you for submitting your manuscript to PLOS ONE. After careful consideration, we feel that it has merit but does not fully meet PLOS ONE’s publication criteria as it currently stands. Therefore, we invite you to submit a revised version of the manuscript that addresses the points raised during the review process.

We look forward to receiving your revised manuscript.

Kind regards,

Piotr Sorokowski

Academic Editor

PLOS ONE

Journal Requirements:

2. Thank you for including your ethics statement:  "This research was conducted under Texas A&M University IRB Protocol IRB2019-0465D. Oral consent was obtained at a community meeting and individually."  

Please amend your current ethics statement to confirm that your named institutional review board or ethics committee specifically approved this study.

Reviewers' comments:

Reviewer's Responses to Questions

**Comments to the Author**

1. Is the manuscript technically sound, and do the data support the conclusions?

Reviewer #1: Yes

Reviewer #2: Partly

2. Has the statistical analysis been performed appropriately and rigorously? 

Reviewer #1: I Don't Know

Reviewer #2: No

3. Have the authors made all data underlying the findings in their manuscript fully available?

Reviewer #1: Yes

Reviewer #2: Yes

4. Is the manuscript presented in an intelligible fashion and written in standard English?

Reviewer #1: Yes

Reviewer #2: Yes

5. Review Comments to the Author

Reviewer #1: Small-Scale Utilitarianism: High Acceptance of Utilitarian Solutions to Trolley Problems among a Horticultural Population in Nicaragua

This study investigates moral decision making using the trolley problem in the Mayangna/Miskito Community. This study examines moral decision making in a less studied population (non-WEIRD). This study and others such as Sorokowski et al. with the Yali community in Papua New Guinea are very welcome contributions to this area of research that examines moral decision making in specific cultural contexts.

The Trolley Problem has been studied extensively even though it has an odd design and uses very strange scenarios. Typically, from a western perspective, participants tend to select the switch/pull the lever option but not to the same extent the push the large person option to save five. Also, as an aside, they are more likely to select this latter option if using a foreign language rather than native language. The authors draw the reader’s attention to three different biases – contact bias (bias against physical act of pushing), means bias (large person serves as the means) and omission bias (to not act). The authors utilise this framework to analyse people’s responses to these scenarios. First, I recommend that the authors explicitly explain these biases in more detail on p. 3 so that a more general audience reader understands the implications of these biases in decision making in these types of scenarios.

The authors also use a content analysis framework for eliciting participant’s explanations for their moral judgment decisions. This is commendable particularly in these types of specifically focused cultural studies.

The Mayangna/Miskito participants in this study are more familiar with death and have high mortality rates in infants and young children. Thus, they are acquainted with illness and death in comparison to more wealthy or industrialised societies. They have also been involved in conflict and combat. I would imagine the original trolley/footbridge dilemmas would be considered odd and unfamiliar whereas the adapted stamped of cattle and additional scenarios would be more appropriate and acceptable for use in this study. In the second study, a Likert scale was used to collect rating data on the acceptability of various scenarios.

Discussion.

I recommend that the Discussion section starts with a more general introductory sentence(s) and refer to the aims of the study. Then proceed to address those aims in a systematic manner to ensure that it is clearer and more reader-friendly or easy to follow.

In sum, according to the authors, Mayayngna/Miskito participants tend to exhibit a contact/means bias, but not an omission bias and are more accepting of utilitarian responses. The authors explain these results in terms of this community living in a harsh environment and being more familiar with illnesses, injuries, violence and higher mortality rates. This makes an interesting comparison with the Yali cultural study. It is important to focus on specific cultural experiences and belief systems when investigating moral decision making rather than using too broad a brush across cultures.

References in the reference list needs revision to comply consistently with referencing conventions.

Reviewer #2: The paper reports two studies investigating moral decision making in remote horticultural population. In contrast to our previous research, the authors found large proportions of participants endorsing the utilitarian solution, whilst showing no sign of the widely reported biases (e.g., the difference between personal and impersonal sacrifice).

Below I provide several critical comments on paper, specifically on the theoretical set up of the studies (or, rather, lack of thereof) and on the design of the studies. To be clear, none of them comments on problems that preclude the publication of the findings but could be solved in a revision.

The introduction lacks a clear rationale for the study other than simply running a study in a novel population (which is acceptable but could be better). A bit more about why variability in moral responding is an issue, and what it means for moral psychology would strengthen the introduction a bit. The dual process framework could be an avenue: it expects higher order cognition to drive U responding, and intuition to drive D responding. Little education would predict more D responses in societies with less formal education.

It was not clear why the trolley vs stampede scenarios were used instead of only the culturally adapted versions. The authors acknowledge that their participants had problems understanding the trolley versions. Moreover, the potential difference test between trolley vs stampede versions is not reported. Next, compared to the older preprint (which I had the opportunity to read few months ago on psyarxiv) includes the ordering effect analysis. It shows that being responded first has an effect on the likelihood of the utilitarian response (not clear how it was estimated in the SEM, since the DV is binary, and the B is > 1!; I assume it is an error). The real issue with ordering effects is more complex. Alex Wiegmann showed multiply times that if the order is indirect -> direct dilemma, respondents show no ordering effect. Yet, having responded to direct dilemma first, participants better realize their action is causing the death of one person, and therefore tend to be less utilitarian is subsequent indirect dilemma. So, it is not enough to look at the relative position of a dilemma, but also at the type of preceding dilemma. Maybe an LME with order, dilemma type (trolley vs stampede), dilemma version (direct vs indirect) would provide more direct answers.

Next, gender effects can be of interest here. I looked at the raw data provided by the authors, and saw that female are far less utilitarian than males. Maybe it would be helpful to additionally control for sex effects in the test above? These can contaminate the main analysis, assuming genders are not perfectly distributed across conditions.

Study 2 is much improved, yet suffers some of the old, and some new issues. Again, the ordering effects are real problem, and can severely affect the responding pattern. Could you supplement the main analysis with one that tests only the first trials for each participant? Next, participants instead of judging the action (act vs does not act) judged the 3rd party that acted utilitarian. Such process is not only a test of moral preferences, but also requires theory of mind, and places the status quo on action: to criticize the sacrifice one must challenge the status quo. Finally, Bostyn in Psych Sci 2018 and PB&R 2019 used such formulation of question to only investigate utilitarian inclinations, claiming that the answer to a Q judging an agent that acted deontologically to test D inclinations. My five, yet unpublished experiments fully support this idea. So, Study 2 tested utilitarian preferences only.

Again, the analysis would be much more sensitive if it would adopt LME modelling instead od series of independent Chi-2 tests. The study would also benefit from clear explanation why for each dilemma, call vs push version was used (to test for the contact vs means bias), and a direct statistical test of this bias. Experiment 2 would also benefit from a figure illustrating its results.

Finally, the discussion is very strong. I liked the framing of our past studies to suggest that deontological actions of Yali are in fact utilitarian, because one prevents further spill of blood in revenge, or because one values own life more than of five other agents (i.e., me + 1 person > 5 persons) making the non-action increasing subjective utility (see Cohen and Ahn, JEP:Gen, 2016).

For minor comments:

Line 287: a part of a sentence is repeated, “and a power of 0.9 to detect a medium effect size of Cohen’s d=0.6”.

Line 315: Although the Trolley scenarios only reveal a suggestive effect (Trolley Switch: 93.5%, Trolley Push: 78.1%, Fisher’s Exact, p=0.148). The effect is either there or not. I don’t know what an suggestive effect with p = > .10 is in statistical terms.

Line 344-346: The authors write: In a robustness check, we examined the possibility of order effects, and although this supplementary analysis provided evidence for a moderate effect, the inferences for the comparisons between the scenarios remain consistent (Electronical Supplemental Materials, Tables S2a–d). I don’t see how ESM supports this claim, please elaborate.

Overall, the paper adds quite a bit to the literature. I think a bot clearer expression of its contribution would strengthen the manuscript. I leave it to the editor and the authors to judge whether my suggestions were helpful, and to challenge any of them if they see it fit.

Best,

Michał Białek

6. PLOS authors have the option to publish the peer review history of their article (what does this mean?). If published, this will include your full peer review and any attached files.

Reviewer #1: **Yes: **Heather Winskel

Reviewer #2: **Yes: **Michał Białek

---

## [Author Response · Author response to Decision Letter 0]

2 Feb 2021

Please find the attached file titled "Response to Reviewers".

---

## [Decision Letter · Decision Letter 1]

17 Mar 2021

Small-Scale Utilitarianism: High Acceptance of Utilitarian Solutions to Trolley Problems among a Horticultural Population in Nicaragua

PONE-D-20-35491R1

Dear Dr. Winking,

We’re pleased to inform you that your manuscript has been judged scientifically suitable for publication and will be formally accepted for publication once it meets all outstanding technical requirements.

Kind regards,

Piotr Sorokowski

Academic Editor

PLOS ONE

Additional Editor Comments (optional):

Reviewers' comments:

Reviewer's Responses to Questions

**Comments to the Author**

1. If the authors have adequately addressed your comments raised in a previous round of review and you feel that this manuscript is now acceptable for publication, you may indicate that here to bypass the “Comments to the Author” section, enter your conflict of interest statement in the “Confidential to Editor” section, and submit your "Accept" recommendation.

Reviewer #1: All comments have been addressed

Reviewer #2: All comments have been addressed

2. Is the manuscript technically sound, and do the data support the conclusions?

Reviewer #1: Yes

Reviewer #2: Yes

3. Has the statistical analysis been performed appropriately and rigorously? 

Reviewer #1: Yes

Reviewer #2: Yes

4. Have the authors made all data underlying the findings in their manuscript fully available?

Reviewer #1: Yes

Reviewer #2: Yes

5. Is the manuscript presented in an intelligible fashion and written in standard English?

Reviewer #1: Yes

Reviewer #2: Yes

6. Review Comments to the Author

Reviewer #1: The manuscript is much improved and has addressed the concerns or comments raised. It is now ready for publication in this special issue.

Reviewer #2: All my comments have been addressed. Thank you for your effort and I hope this paper will be cited a lot!

7. PLOS authors have the option to publish the peer review history of their article (what does this mean?). If published, this will include your full peer review and any attached files.

Reviewer #1: **Yes: **Heather Winskel

Reviewer #2: **Yes: **Michał Białek

---

## [Editor Report · Acceptance letter]

23 Mar 2021

PONE-D-20-35491R1 

Small-Scale Utilitarianism: High Acceptance of Utilitarian Solutions to Trolley Problems among a Horticultural Population in Nicaragua 

Dear Dr. Winking:

I'm pleased to inform you that your manuscript has been deemed suitable for publication in PLOS ONE. Congratulations! Your manuscript is now with our production department. 

Kind regards, 

on behalf of

Dr. Piotr Sorokowski 

Academic Editor

PLOS ONE